# Real-World Safety and Efficacy Clinical Data of an Improved Allergen-Specific Immunotherapy Product for the Treatment of Bee Venom Allergy

**DOI:** 10.3390/vaccines11050979

**Published:** 2023-05-13

**Authors:** Luis Alfredo Gonzalez Guzman, Jose Carlos García Robaina, Javier Barrios Recio, Elena Escudero Arias, Tania Liñares Mata, Raquel Cervera Aznar, Federico De La Roca Pinzón, Lissette del Carmen Miguel Polo, Luis Arenas Villarroel, Verónica P. López Couso, Javier Alcover Diaz, David Rodriguez Gil, Ricardo Palacios Pelaez, Francisco Javier Carballada Gonzalez

**Affiliations:** 1Allergy Department, Hospital Lucus Augusti, 27003 Lugo, Spain; 2Allergy Department, Nuestra Señora de la Candelaria University Hospital, 38010 Santa Cruz de Tenerife, Spain; 3Allergy Department, Hospital Provincial de Pontevedra, 36001 Pontevedra, Spain; 4Allergy Department, Hospital General Universitari de Castelló, 12004 Castelló de la Plana, Spain; 5Allergy Department, Hospital Clínic de Barcelona, 08036 Barcelona, Spain; 6Allergy Department, Hospital Virgen De La Luz, 16002 Cuenca, Spain; 7Allergy Department, Hospital do Meixoeiro, 36214 Vigo, Spain; luisarenas.alergia@gmail.com; 8Laboratorios de Diagnóstico y Aplicaciones (Diater), 28919 Madrid, Spain

**Keywords:** hymenoptera venom, allergy immunotherapy, Api m 10, bee venom, human serum albumin, vaccines

## Abstract

The aim of this study is to explore the safety and efficacy of bee venom immunotherapy without HSA, in real-life patients. Methods: This is an observational retrospective study developed in seven hospitals in Spain, where patients treated with this immunotherapy were included. They gathered the protocol used to initiate the immunotherapy, adverse reactions, field re-stings, and the patient clinical data (clinical history, biomarkers, and skin prick test). Results: A total of 108 patients were included. In total, 4 protocols were used (5 weeks reaching 200 μg, and 4, 3, and 2 weeks reaching 100 μg). An incidence of systemic adverse reactions for each 100 injections of 1.5, 1.7, 0, and 0.58, respectively, was found. The demographic data showed not to directly affect the appearance of adverse reactions, except for those having a grade 2 systemic reaction with immunotherapy previously had a grade 4 systemic reaction; the IgE to Apis mellifera was 3 times higher in patients with systemic reactions of grade 1 than in the general group, and other specific IgEs were lower in those with systemic reactions. Most of the patients recognized Api m 1 followed by Api m 10. In the sample, 32% experienced spontaneous re-stings, without presenting systemic reactions, after a year of treatment.

## 1. Introduction

Hymenoptera venom immunotherapy is the only curative treatment for those sensitized patients that experienced systemic reactions to hymenoptera stings [1,2,3].

Allergy to honeybee venom (HBV) among beekeepers is around 36%, while in the general population the prevalence of HBV allergy is around 15–25%, with a mortality rate due to hymenoptera stings that varied from 0.02 to 0.19 per million inhabitants in Spain [4,5,6,7,8].

Honeybee venom immunotherapy (HBVIT) has demonstrated to be effective for these patients but is highly associated with adverse effects that can affect up to 40% of patients with HBVIT, compared to wasp venom immunotherapy which is better tolerated [9,10,11]. 

Due to these adverse reactions during immunotherapy, the buildup phases with HBVIT were very long (8–12 weeks); nowadays, thanks to the improvement in the extracts and knowledge of the disease, the appearance of adverse reactions is decreasing, so practitioners are trying to reduce the buildup phases in favor of patient comfort [12,13,14,15].

The allergen content of these immunotherapy extracts might differ due to natural variations in the source material or different processing strategies of the manufacturers, affecting the effectiveness of the immunotherapy [16]. 

Human Seric Albumin (HSA) is incorporated in vaccines to stabilize the extract and to prevent immunogens from adhering to the walls of the injection vial [17,18]. HSA is widely used in venom extracts, and although it covers some of the allergens, making it difficult to distinguish all of them, it stabilizes the extract, making it last longer. 

The aim of this study is to explore the real practice in Spain among the hospitals that use this improved bee venom immunotherapy to seek the safety of this product, studying the overall safety and the association between the demographic data of the patients and the appearance of adverse reactions.

## 2. Materials and Methods

This study was an observational retrospective study conducted in seven hospitals in Spain. Its principal objective was to evaluate the safety of immunotherapy with bee venom (*Apis mellifera*) without HSA (Diat Venom Apis). This study was approved by the Comité de ética de la investigación con medicamentos de Galicia (CEIm-G)) in 2020, with registration code 2020/553. 

### 2.1. Patients

The study population was selected from seven hospitals in Spain, which were in Pontevedra, Vigo, Lugo, Castellón, Cuenca, Tenerife, and Barcelona; the selection of patients was made from the clinical history, selecting those who have been treated with Diat venom apis after it was released, older than 18 years old, and have not undergone HBVIT in the previous five years. (The PI and the recruitment per hospital can be found in Appendix A Table A1)

The following data were required from the clinical history of the selected patients:Demographic data of the patients (age, gender, profession, and clinical history) (Table 1).Initial reaction to the bee field sting (Müller grades) (Table 1).Immunotherapy initiation protocol used. Four different protocols were used in the study, which are shown in Table 2, grouped according to the weeks employed in the buildup phase.Skin prick test to *Apis mellifera*.Total IgE, tryptase, and specific IgE to *Apis mellifera* and the different molecular components upon each hospital’s availability. IgEs were measured by InmunoCAPTM (Table 3).Adverse reactions during the immunotherapy and its classification.Spontaneous re-sting after initiating the immunotherapy.

### 2.2. Venom Extract

Diat venom apis is a whole venom extract made of national ecological bees from the northern province of Leon, Spain. The collection method is by electrical stimulation, a technique widely used that allows us to collect honeybee venom without killing them or changing their habits, although this produces an increase in bee aggressiveness during the activation time of the device. In this case, it was 45 min in different cycles, and afterward the bees returned to normal. These bees were supplemented when feeding so they did not have to move far and could stay along the hives. The lyophilization of the product takes place in less than 48 h from the collection, after it is processed by purification and manitol added. The final product is a lyophilized monodose extract that has to be reconstituted with 1.2 mL of saline, obtaining a dose of 120 µg of bee venom immunotherapy. 

The characterization of the venom was performed with the combination of two techniques. In gel-digestion approach, proteins separated on SDS-PAGE gel were digested. To increase the depth characterization of venom extract, a direct in-solution trypsin digestion of proteins was performed, and peptides were identified by LC-MS. This approach also allows to estimate the relative abundances of the proteins identified by shotgun proteomics.

In gel-digestion approach, bands resolved were identified. Api m 1 (30.00%), Api m 2 (1.37%), Api m 3 (0.17%), Api m 4 (35.34%), Api m 5 (1.47%), Api m 6 (3.87%), Api m 7 (2.41%), Api m 8 (0.23%) Api m 9 (0.63%), Api m 10 (1.26%), and Api m 11 (1.50%) were identified in solution-digestion approach, in venom extract. Due to technique limitations, Api m 12 was not identified. 

The combination of both strategies allows to identify the component-resolved diagnosis and other relevant allergens in honeybee venom extract, reported by IUIS database [19].

### 2.3. Statistical Analysis 

Categoric variables are described by absolute and relative frequencies. Chi-square test or Fisher test were used to evaluate the relation between categoric variables.

To describe continuous variables, the mean and standard deviation were used. To compare two groups of patients with quantitative variables, Student’s *t*-test was used for the parametric tests, and for non-parametric tests, Mann–Whitney U was used. To compare three groups of patients, ANOVA was used. 

The incidence of AR for each 100 injections was estimated by an extrapolation of the coefficient of the total number of AR and the total number of injections administered during the study. 

The used significance level of statistics is 0.05. All the data were analyzed with SAS version 9.4.

## 3. Results

In total, 108 patients were included in the study. One of those patients had to be discarded because they did not meet all the inclusion criteria. So, finally, 107 patients were recruited. 

### 3.1. Safety of the Immunotherapy 

#### 3.1.1. Adverse Reactions during Immunotherapy

In the study sample, 18 patients (16.8%) experienced an adverse reaction (AR); a total of 21 adverse reactions (19.6%) were reported in these 18 patients. There were 13 local reactions (12.15%) and 8 systemic reactions (7.48%); among the systemic reactions, 4.7% were grade 1 and 2.8% were grade 2. All systemic reactions were immediate reactions, but among the local reactions, the majority (9, 8.4%) were delayed reactions. 

In Table 4 we see the number and percentage of adverse reactions per protocol. 

Among the four different protocols, an incidence of systemic adverse reactions was found for each 100 injections of 1.5, 1.7, 0, and 0.58, respectively, for protocols 1, 2, 3, and 4.

In relation with the time of appearance of the systemic AR, in the four protocols the AR appeared mostly in the first week, with most of the ARs after the first dose (10 µg), except for protocol 2 where the majority were in the third week (50 µg).

All systemic reactions were resolved immediately at the immunotherapy unit and no patient required hospitalization or emergency care. 

Systemic grade 1 reactions involved generalized pruritus or urticaria; systemic 2 reactions were nausea and dizziness. 

One patient was taken off the treatment in protocol 1 because he had two systemic reactions (generalized pruritus), one with 10 µg and with 5 µg.

The patients in protocol 4 (that reached 2 µg) only had ARs during the first buildup dose with 10 µg while having a good tolerance with higher doses.

#### 3.1.2. Safety Associated to Demographic Data and Sensitization Profiles

Table 5, shows the correlation of the demographic data of the patients with the presence of ARs. There is only one statistically significant association, and it is that the patients that had a grade 4 reaction with the field sting had a grade 2 reaction with HBVIT. 

In this sample, 41 patients had a history of previous diseases, most of them cardiovascular disease, such as hypertension (23%), heart attack (6%), or atrial fibrillation (3%), among others. Meanwhile, others had allergic rhinitis, urticaria, drug allergy (7%), mastocytosis (6%), tumors (1%), or an autoimmune disease, such as psoriasis or hypothyroidism (4%). 

Among the patients that had systemic AR, 66.6% had previous disease, 2 had mastocytosis, and another 2 had had atrial fibrillation and heart attack.

The total IgE and specific IgE that were available for each patient were collected and the ratio between the specific IgE and total IgE was calculated, as is described in the INTEGRA published papers [20,21] shown in Table 6. 

All patients recognized sIgE to *Apis mellifera*, and Table 7 shows how many patients were sensitized to each allergen in the available data. The predominant allergen recognition was to Api m 1 and Api m 10, and the monosensitized population was calculated only among the patients that had all allergens tested. There were 11 patients monosensitized, and in 6 of them (54.5%), it was to Api m 10. The recognition profile rate observed in this study was in order of prevalence of 87%, 86.25%, 48%, 47.6%, and 44% for Api m 1, Api m 10, Api m 2, Api m 5, and Api m 3, respectively.

### 3.2. Efficacy, Field Re-Stings during Immunotherapy

After one year of receiving immunotherapy to BV with Diat Venom Apis, thirty-four patients experienced multiple new field stings, with good tolerance and no systemic reactions, being asymptomatic. There was not any correlation between having ARs during immunotherapy and having reactions with a new sting or vice versa. 

The sensitization profile was diverse, and all the patients recognized the *Apis mellifera* whole extract. After it, the most recognized was Api m 1, and in order of prevalence, it was Api m 10, Api m 2, and Api m 5, with Api m 3 last. There were four patients sensitized to all allergens. There were none monosensitized. The summarized profile is shown in Table 8. 

In Appendix A Table A3, there is the detailed sensitization profile of these patients. 

## 4. Discussion

Bee venom immunotherapy is the recommended and only curative treatment for patients allergic to honeybee venom that develop systemic reactions after their sting [4,7,16]. 

The HBVIT extract is the key for successful treatment. Over the years, commercialized extracts always had HSA [16], which has both benefits and disadvantages; some benefits are, for example, that the vial can be used for multidoses, with better conservation, but a major disadvantage is that the presence of all allergens cannot be assured when analyzing because HSA covers some of them. This extract was studied by proteomic identification, using two techniques [22], showing that it is the most reliable compared to raw venom, which is important because patients allergic to honeybee venom may change their sensitization depending on the area to which they are exposed [16]. Usually, Api m 1 is the majority, but there are some patients monosensitized to Api m 10 or a high prevalence of Api m 6 in some regions of Spain [22], so the effectiveness of the immunotherapy is totally dependent on the type of extracts. In some studies [23,24], it has been observed that immunotherapy fails in patients sensitized or monosensitized to Api m 10; so, it is very important for all allergens to be present in the extract. 

In this study, the majority (54.5%) of monosensitized patients were sensitized to Api m 10 and had a good tolerance not only to the immunotherapy but also to a field re-sting. One of them reached a maintenance dose of 200 µg, while the others were treated with the standard concentration of 100 µg. According to the previously mentioned studies [23], if Api m 10 is present in the extract, we can achieve a good treatment for the patient. There were only 10 patients sensitized to all allergens. The recognition profile ratio is similar to other studies with Api m 1, Api m 2, and Api m 3, where the rate varies from 60–97% for Api m 1, 46–52% for Api m 2, and 38–50% for Api m 3. However, for Api m 5, there was a higher ratio in other studies (58–60%), and for Api m 10, there was a higher ratio in this study compared to others, where the ratio is between 49 and 62% for Api m 10 [25]. In this study, there is a higher prevalence of patients sensitized to Api m 10, noting the importance of high-quality extracts with the presence of all allergens, and the difference among other populations around Europe.

HBVIT has always been related to a higher number of ARs than other hymenoptera venom immunotherapy [11,13]. In the literature, an incidence of adverse reactions was found that varies from 0% to 46% [26], while recent studies [12] have published an incidence of adverse reactions of 28.9% [26] with HBV, with a frequency of 50.4% for systemic reactions and 10% for large local reactions. In this study, there was an incidence of ARs of 19.6%, of which 7.5% were systemic reactions not greater than grade 2. The buildup phase is the riskiest moment for having an AR, and nowadays, faster protocols are associated with fewer ARs than the long ones [15,27]. There is a study in which they observed that in patients who did not tolerate the updosing with conventional protocols, going for a faster buildup made them tolerate and reach the maintenance dose without ARs [28]. The safest protocol in this study was protocol 3 (two weeks), with no systemic reactions and an incidence of systemic reactions of 0 for each 100 injections. Above all, all the adverse events were immediately resolved with no consequences; only one patient had to stop the immunotherapy because he had two systemic reactions (grade 1) with two different doses (0.05 mL and 0.1 mL) in protocol 1 (four weeks). 

In a study of 143 patients comparing 3 different protocols with 54 HBVIT and 93 wasp VIT, they found a presence of systemic reactions among the 3 protocols (cluster 3 weeks, 3-day rush, and half-day ultra rush) of 25%, 8.7%, and 15.8% to HBVIT and 12.5%, 0%, and 6.1% for wasp VIT, respectively [27]. In this study the incidence is 1.9%, 4.7%, 0%, and 0.9% for each protocol, with a higher study cohort. Most of the ARs were with doses that varied from 10 µg to 50 µg, with most of them occurring during the first week of administration. None of the patients in protocol 4 had any reaction in the maintenance phase, and all of them tolerated well 200 µg during the study period. According to what has been observed in previous studies, the riskiest phase is when updosing [13,29].

There are some risk factors related to the appearance of ARs during immunotherapy, which are the following [5,14]: hypertension, mastocytosis, β-blockers/ACE inhibitors intake, serum basal tryptase concentration, male sex, old age, etc. [13,29]. In the demographic data, all these items were taken into account to see if there was a correlation with the severity and form of the AR. In general, any statistical difference between all the parameters observed was found, as can be seen in Table 5. The age of the patients that presented with an AR was around 50 years old, being the same mean age as those that did not have any ARs, while there was a slight increase in those with a systemic reaction grade 2 up to 63 years old, but this is not statistically significant. Meanwhile, patients with local reactions had a mean age around 47 years old. It is usually believed that the older the patient, there are more probabilities they will have a reaction [30]. Between beekeepers and non-beekeepers, there was a ratio of 1:2 of patients with reactions, respectively, which is also not statistically significant. Regarding the grade of the initial reaction with no statistical difference, except for the patients that suffered grade 2 ARs, all of them (2) had a grade 4 initial reaction (*p* = 0.0129), while the risk of having another systemic reaction after already having had one was around 70% in adults and 30% in children, even 10 to 20 years after the reaction [10]. 

The relationship between the total and specific IgE and basal tryptase to the appearance of ARs was studied; it was found that the total IgE was lower in those patients that experienced an AR than in the rest of the patients. The ratio of IgEs/IgEt [20,21] was calculated for each allergen so as to correlate them among the groups: the IgE ratio to *Apis mellifera* was similar to all the groups, except for the patients that had a systemic reaction of grade 1 that had an IgEs/IgEt ratio of 39 instead of the mean of 10; the Api m 1 ratio was 4 times lower in patients with systemic reactions than in those with no reactions, which was similar to Api m 2; with Api m 10, there was a 3-fold decrease in those patients with systemic reactions; and Api m 3 and Api m 5 were similar among the groups. In other studies, they found that sIgE to hymenoptera venom was correlated to the severity of the reactions during the VIT buildup phase, but that is controversial nowadays. In this study, it was found that the higher sensitization to the different molecular diagnostics was not correlated to a higher presence of ARs [29,31,32], except for IgE *Apis mellifera*; basal serum tryptase was a major factor for developing systemic reactions correlated with age and male sex, [31,33]. In addition, despite the fact that in this study this biomarker was not statistically different, there was a slight increase in those patients that experienced a systemic reaction not related to any other variable, as most of the studies suggest [13,29,33]. In recent studies, they recommend increasing the maintenance dose in these at-risk patients to 200 µg or even maintain a life-long 200 µg dose every 12 weeks to protect these patients [34]. Another studied biomarker was CCDs sensitization which is known to be important in cross reactivity, especially in alcohol-drinking patients [35,36], but is seen to be low in beekeepers. In this study, 51 patients were tested for CCDs, and it was observed that beekeepers had a higher sensitization to CCDs than non-beekeepers (*p* = 0.038) but not related to a higher frequency of ARs. 

During the study, 34 patients experienced multiple field stings; none of them had a systemic reaction. These re-stings are a valuable test to see the effectivity of the immunotherapy, considering that these patients only had had one year of immunotherapy and they already tolerated bee stings. No relation was found among the demographic data of the patients, or the severity of the first sting, or ARs during HBVIT. Among the patients that were re-stung, there were patients that had an initial reaction of grade 4 and now no reaction to the new sting after a one-year treatment. In some studies, after the first year of treatment they have found an incidence of 80% of local reactions after re-stings [37,38], with about 80% of patients experiencing local reactions during VIT. In this study, only 10% had local reactions after the re-sting and 0% had systemic reactions. 

So, after the first year of treatment, some patients have tolerated new bee stings and could continue with their normal daily activities, which implies a cost benefit of this immunotherapy, reducing emergency visits and pharmacological costs [39]. 

## 5. Conclusions

In conclusion, HBVIT is the only curative treatment for patients allergic to honeybee venom and it is essential to have a complete extract to treat all those patients regardless of their sensitization. The safety and indicating efficacy data support that this new honeybee venom extract has a balanced safety and efficacy profile indicated to treat bee venom allergic patients with different sensitization profiles. More studies will be needed to prove the disease-modifying potential of the product.

## Figures and Tables

**Table 1 vaccines-11-00979-t001:** Demographic data of the study population.

Demographic Data					
Age (years)					
Mean (SD)	52.12 (14.63)				
95% CI	(49.32; 54.93)				
Mediana (P25; P75)	53 (41; 62)				
(Min; Max)	(20; 85)				
N	107				
Sex N (%)	107 (100%)				
Men	76 (71%)				
Women	31 (29%)				
Profession ^1^ n (%)	105 (100%)				
Beekeeper	55 (52.38%)				
Other	50 (47.7%)				
Previous diseases ^2^	107 (100%)				
Yes	41 (38.3%)				
No	66 (61.7%)				
Initial field-sting reaction	107 (100%)				
Local	0 (0%)				
Systemic	107 (100%)				
Immediate (<one hour)	107 (100%)				
Late onset (>one hour)	0 (0%)				
Grade of systemic reaction (Müller grades)	N	Grade 1	Grade 2	Grade 3	Grade 4
	107 (100%)	23 (21.5%)	42 (39.2%)	25 (23.4%)	17 (15.9%)

^1^ N = 2 patients whose profession was unknown. ^2^ Previous diseases are described in Appendix A Table A2.

**Table 2 vaccines-11-00979-t002:** Protocols employed.

Schedule				Patient per Protocol
		Dose, mL	Dose, µg	
Protocol 1	Week 1	0.1 mL	10 µg	22 patients
	Week 2	0.2 + 0.3 mL	20 µg + 30 µg	
	Week 3	0.5 + 0.5 mL	50 µg + 50 µg	
	Week 4	1 mL	100 µg	
Protocol 2	Week 1	0.1 + 0.1 mL	10 µg + 20 µg	48 patients
	Week 2	0.2 + 0.3 mL	20 µg + 30 µg	
	Week 3	0.5 + 0.5 mL	50 µg + 50 µg	
Protocol 3	Week 1	0.1 + 0.2 + 0.2 mL	10 µg + 10 µg + 20 µg	18 patients
	Week 2	0.5 + 0.5 mL	50 µg + 50 µg	
Protocol 4	Week 1	0.1 + 0.1 mL	10 µg + 10 µg	19 patients
	Week 2	0.2 + 0.3 mL	20 µg + 30 µg	
	Week 3	0.5 + 0.5 mL	50 µg + 50 µg	
	Week 4	0.75 + 0.75 mL	75 µg + 75 µg	
	Week 5	1 + 1 mL	100 µg + 100 µg	

**Table 3 vaccines-11-00979-t003:** Sensitization profiles previous immunotherapy.

	Total IgE (kUA/L)	IgE *Apis mellifera* (kUA/L)	IgE Api m 1 (kUA/L)	IgE Api m 2 (kUA/L)	IgE Api m 3 (kUA/L)	IgE Api m 5 (kUA/L)	IgE Api m 10 (kUA/L)	Tryptase (g/L)
Mean (SD)	203.32 (520.30)	21.44 (29.38)	9.95 (20.51)	3.36 (13.29)	0.97 (2.99)	4.05 (12.65)	7.12 (13.85)	5.41 (2.84)
95% CI	(102.14; 304.51)	(15.79; 27.10)	(5.68; 14.22)	(0.09; 6.62)	(0.24; 1.71)	(0.87; 7.24)	(4; 10.24)	(4.84; 5.98)
Median (P25; P75)	81.77 (34.50; 194.50)	9.30 (2.44; 24.80)	1.74 (0.34; 6.03)	0.08 (0; 0.88)	0.06 (0; 0.30)	0.10 (0; 2.05)	1.08 (0.34; 7.09)	4.85 (3.60; 6.20)
Min; max	(1.46; 4945)	(0; 100)	(0; 95)	(0; 100)	(0; 16.3)	(0; 83)	(0; 70.80)	(1.80; 19.80)
N	104	106	91	66	66	63	78	90

**Table 4 vaccines-11-00979-t004:** Number of ARs for each protocol.

AR	Protocol 1	Protocol 2	Protocol 3	Protocol 4	Total
	N (%)	N (%)	N (%)	N (%)	N (%)
Local AR	7 (6.5%)	0	4 (3.7%)	2 (1.9%)	13 (12.1%)
Systemic AR	2 (1.9%)	5 (4.7%)	0	1 (0.9%)	8 (7.5%)

**Table 5 vaccines-11-00979-t005:** Demographic data in relationship to the presence of AR.

		General	No AR	Systemic AR	Systemic AR Grade 1	Systemic AR Grade 2
Age (years)		52.12	52.3	57.7	54.75	63.5
*p*-value		0.7496		0.3416	0.7160	0.2688
Gender	Male	71% (76)	82.9% (63)	9.2% (7)	5.3% (4)	4% (3)
	Female	29% (31)	83.9% (26)	3.2% (1)	3.2% (1)	0%
*p*-value		0.9025		0.6699	1.000	1.0000
Profession	Beekeeper	52% (55)	56% (49)	3.6% (2)	1.8% (1)	1.8% (1)
	Non Beekeeper	48% (50)	44% (39)	8% (4)	6% (3)	2% (1)
*p*-value		0.2513		0.5963	0.4654	0.4258
Severity of sting reaction (Müller)	Grade 1	21.5% (23)	20.2% (18)	4.3% (1)	4.3% (1)	0%
	Grade 2	39.3% (42)	39.3% (35)	4.7% (2)	4.7% (2)	0%
	Grade 3	23.4% (25)	27% (24)	4% (1)	4% (1)	0%
	Grade 4	15.9% (17)	13.5% (12)	11.7% (2)	0%	11.7% (2)
*p*-value		0.1536		0.1898	0.8468	0.0129

**Table 6 vaccines-11-00979-t006:** Total and specific IgE in relation to the presence of AR. (NA: Not applicable).

		Mean/Ratio ^¶^			
	General	No AR	Systemic AR	Grade 1	Grade 2
IgE total (kU/L)	203.32/NA	226/NA	102.4/NA	81.52/NA	144.3/NA
IgE *Apis mellifera* (kU/L)	21.4/10.5	23.1/10.2	26.5/11.7	31.78/39	15.98/11
IgE Api m 1 (kU/L)	9.95/4.9	10.8/4.8	0.83/0.8	0.83/1.01	-
IgE Api m 2 (kU/L)	3.36/1.65	3.8/1.7	0.22/0.2	0.22/0.27	-
IgE Api m 3 (kU/L)	0.97/0.48	1.1/0.48	0.18/0.18	0.18/0.22	-
IgE Api m 5 (kU/L)	4.05/2	4.46/2	1.89/1.8	1.89/2.3	-
IgE Api m 10 (kU/L)	7.12/3.5	7.86/3.5	0.94/0.9	0.94/1.15	-
CCD’s (kU/L)	0.48/NA	0.48/NA	0/NA	0/NA	-/NA
Triptasa (g/L)	5.41/NA	5.33/NA	7.49/NA	8.82/NA	2.2/NA

^¶^ Ratio has been calculated; component-sIgE/whole extract-sIgE following INTEGRA [20] instructions. AR: adverse reactions. NA: not applicable.

**Table 7 vaccines-11-00979-t007:** Sensitization profile in this study population with the available data.

sIgE	*Apis mellifera*	Api m 1	Api m 2	Api m 3	Api m 5	Api m 10	CCD’s
N total	107	93	68	66	65	80	51
Positive	107	82	33	29	31	69	17
Monosensitized		2	3	0	1	6	0

**Table 8 vaccines-11-00979-t008:** Sensitization profile of the patients that had a field sting during immunotherapy.

	Total IgE (kUA/L)	IgE *Apis mellifera* (kUA/L)	IgE Api m 1 (kUA/L)	IgE Api m 2 (kUA/L)	IgE Api m 3 (kUA/L)	IgE Api m 5 (kUA/L)	IgE Api m 10 (kUA/L)	Tryptase (g/L)
Mean (SD)	135.35 (212.05)	23.94 (40.53)	12.23 (23.55)	2.53 (6.76)	0.33 (1.14)	2.34 (4.62)	6.59 (14.5)	5.14 (2.43)
Min; max	(1.46; 4945)	(0.25; 100)	(0.07; 95)	(0; 30.9)	(0; 5.88)	(0; 21.9)	(0.01; 70.80)	(1.9; 14.7)
Percentage of sensitization	NA	100%	100%	65.4%	53.85%	65.4%	100%	NA
N	34	34	29	26	26	26	26	31

## Data Availability

The authors ensure that data shared are in accordance with consent provided by participants on the use of confidential data, following Reglamento (UE) 2016/679 del Parlamento europeo y del Consejo de 27 de abril de 2016 de Protección de Datos (RGPD) and the good clinical practice and declaration of Helsinki. The data presented in this study are available on request from the corresponding author. The data are not publicly available due to GDPR.

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
