# Peer review of "Real-World Safety and Efficacy Clinical Data of an Improved Allergen-Specific Immunotherapy Product for the Treatment of Bee Venom Allergy"

_vaccines, 2023, doi:10.3390/vaccines11050979_

Round 1
Reviewer 1 Report
Real world experience of a multicentric study using bee venom immunotherapy without human serum albumin in Spain; the importance of molecular characterization.
By Luis Alfredo Gonzalez Guzman et al.
Esteemed Authors,
I have reviewed your manuscript. I find it of interest but cannot accept it in its current form for publication in Vaccines.
A general comment on the form of the manuscript:
I recommend that the team of authors take the necessary time in its preparation. It must be adapted to the specific style of Vaccines, for example regarding the list of references to how they are cited in the body of the manuscript. It must be corrected. In general, the text has a lot of room for improvement. It should be structured in paragraphs (not in sentences). The linguistic style used is very difficult to understand.
Title:
How research is described, particularly in a title, is important because it influences how the paper describing the study is retrieved, and also suggests how the study should be analysed and reported. From this point the title must be rewritten. I think that the expressions “real world experience” and “molecular characterization” should be avoided.
Abstract: The aim of this study is to explore the safety and efficacy of bee venom immunotherapy without HSA, in real life patients. In other parts of the MS the authors use: “This is an observational retrospective study”; “Real world experience of a multicentric study”; Lines56-57: “The aim of this study is to explore the real practice in Spain among the hospitals that use bee venom immunotherapy without HSA to seek the safety of this product." I think it is better clearly to define the class of study that was carried out... a pilot or a randomized bee venom immunotherapy study.
Introduction
L37-38
“Allergy to honeybee venom (HBV) among beekeeper it’s around 36%; while in general population the prevalence of HBV allergy is around 15-25%, with a mortality rate about 0.008 per million of inhabitants in Spain [4] [5] [6] [7].”
I checked the above references (4 to 7) and it was not possible to find/support the statement of a 0.008 mortality rate in Spain due to honeybee stings. Could you provide the source?
L52-55
Authors say: Human Seric Albumin (HSA) is incorporated in vaccines to stabilize the extract and to prevent immunogens from adhering to the walls of the injection vial. However, because HSA is a protein derived from human blood, there is a theoretical risk that it might contain infectious agents which could make it unsafe for humans. [16]
But if you check reference 16, we can read as the conclusion of the above paper that: human serum albumin … in vaccines has not been found to be harmful in humans or experimental animals.
M and M
Need to be clarified. Table 1 should be present here and information provided in Results (Lines 90-95). Table 1 is not well formatted, so it is difficult to understand each of the protocols used.
Any information about the characteristics of the Diat Venom Apis needs to be presented here.
The patient inclusion and exclusion criteria should be clearly mentioned in M & M.
Were subjects excluded if they had previously undergone honeybee VIT?. What about the geographical distribution of the participants?
Results
The results section is divided into 3 parts: primary objective, secondary objectives and field stings. I miss to define clearly what the above objectives are, in M & and M. Moreover Results in the Tables should include the statistical significance where appropriate.
Discusion
Authors say: Lines 150-162
“The HBVIT extract is the key for a successful treatment; through the years, commercialized extracts have always had HSA [15]; this rapports benefits and disadvantages; benefits like, the vial can be used as multidose, better conservation, but a major disadvantage is that it cannot assure the presence of all allergens, because HSA covers some of them while analyzing. “
Regarding the above, it looks like not clear. In the introduction section authors say that the disadvantage of use VIT with HSA is a potential risk with infections agents. Now is suggested that there is a relation between the presence of all allergens because of HSA. The above need to be clarified.
“This extract was studied by proteomic identification, [19] showing that is the most reliable compared to raw venom, having the presence of all allergens. “
When the authors referred to “this extract”, it means Diat Venom Apis?. I checked reference 19, in reference 19 we can read that: In conclusion, despite the possible limitations of this study, such as the nondetection of Api m 12 due to the limitations of the techniques used, The above need to be clarified.
Conclusions
VIT represents an effective treatment against bee venom allergy. However, it involves long treatment times, high costs, and the risk of adverse events. Shorter, safer, and cheaper treatment options are therefore pursued. Has the study carried out here any limitations? Any advantage over the other VIT based in different venom extracts with different adjuvants?
Author Response
Esteemed editor;
First of all thank you for reviewing our manuscript and found it interesting.
General comment:
- We have done an extensive English review by a professional translator enterprise.
- Changed the style of the manuscript so is easier to understand.
Title:
The title has been changed as suggestion for; "Real-world safety and efficacy clinical data of an improved Allergen Specific Immunotherapy product for the treatment of bee venom allergy"
Introduction:
L37-38
"I checked the above references (4 to 7) and it was not possible to find/support the statement of a 0.008 mortality rate in Spain due to honeybee stings. Could you provide the source?"
- The introduction has been modified, inserting reference 6 for the chapter 1 from the same book.
6. V. Soriano Gomis, R. Guspi Bori. Introducción y epidemiología de la alergia al veneno de los himenópteros. Tratado de alergología . 2017.
L52-55
"But if you check reference 16, we can read as the conclusion of the above paper that: human serum albumin … in vaccines has not been found to be harmful in humans or experimental animals."
- Reference 16 changed to reference 17.
Varicella vaccine without human serum albumin versus licensed varicella vaccine in children during the second year of life: a randomized, double-blind, non-inferiority trial. https://doi.o. Prymula, R., Simko, R., Povey, M. et al. 7, 2016, BMC Pediatr, Vol. 16.
M and M
Need to be clarified. Table 1 should be present here and information provided in Results (Lines 90-95). Table 1 is not well formatted, so it is difficult to understand each of the protocols used.
- Table 1 has been added to M and M, and re-styled for a better comprehesion.
Any information about the characteristics of the Diat Venom Apis needs to be presented here.
The patient inclusion and exclusion criteria should be clearly mentioned in M & M.
Were subjects excluded if they had previously undergone honeybee VIT?. What about the geographical distribution of the participants?
- - We have added a detailed parragraph about the characteristics of the venom and the patients data.
Results
"The results section is divided into 3 parts: primary objective, secondary objectives and field stings. I miss to define clearly what the above objectives are, in M & and M. Moreover Results in the Tables should include the statistical significance where appropriate. "
- We have modified the results section in three sections and clearly divided and better explained.
Discusion
Lines 150-162
- We have clarified the both disadvantages of the human serum albumin, at the introduction and at the discusion.
"I checked reference 19, in reference 19 we can read that: In conclusion, despite the possible limitations of this study, such as the nondetection of Api m 12 due to the limitations of the techniques used, The above need to be clarified. "
- This point has been corrected according to the article.
Conclusions
"VIT represents an effective treatment against bee venom allergy. However, it involves long treatment times, high costs, and the risk of adverse events. Shorter, safer, and cheaper treatment options are therefore pursued. Has the study carried out here any limitations? Any advantage over the other VIT based in different venom extracts with different adjuvants? "
- By now the study has the limitations that is an observational retrospective study were we couldn't get all the desirable data, because it was only the one physicians report on their daily practice, we are collecting all economic, sfaety and efficacy data for future studies.
Reviewer 2 Report
The current manuscript present real-life clinical efficacy and safety data of an improved bee venom Immunotherapy product.
In a retrospective analysis of 108 treated patients, safety profile is analyzed , with a detailed stratification based on molecular sensitization profiles.
The data presented are interesting and offer relevant new information. Although the study design is not optimum, for example is not prospective, DBPC…, it is true that in AIT on venom allergy the inclusion of placebo groups is morally questionable and the efficacy data should be evaluated by spontaneous re-stinging or by re-stinging test.
General comments:
1. Grammar. The article is difficult to follow. A complete rewriting by an English speaker is strongly recommended. Check homogeneity of verbal tenses. There are multiple grammar errors along the article.
2. Title. The title should reflect the reality of the study, avoiding “marketing messages”. An alternative title suggestion might be:
“Real-world safety and efficacy clinical data of an improved Allergen Specific Immunotherapy product for the treatment of bee venom allergy”
As the lack of HAS or alum is an improvement of the formulation, that makes the product an improved formula compared to existing products
3. In general when an original article, sponsored by a Company is submitted, there should be an special care to present scientific evidence without a clear marketing orientation. By doing that, the strength of the publication and lack of bias will be increased and at the end it will be even more positive for the Company purpose. I suggest to revise
Specific comments:
1. Introduction: Please describe in detail the aim of the study, aligning this part with the results (Primary objective, secondary …..). Avoid the “without HSA” along the manuscript as is a quality characteristic of the product, not connected as such with the clinical outcomes.
2. Materials and Methods. Please describe in detail the Diat Venom Apis product. Is a whole venom extract? How is formulated? Is a Freeze-dried presentation? How is reconstituted? What is the recommended dosing? Has been already described in previous publications in detail? What is the allergenic composition? For example levels of Api mi1, Api m 10…. The potential advantages of the product compared to others should be described here. I understand that are related to a better potential safety profile (lack of human derivatives, lack of adjucants) without compromising stability and clinical benefit.
3. A demographic description of the 107 patients included could be incorporated in M&M section describing mean values and SD of the different variables upon inclusion.
4. The different protocols used in AIT should be part of the M&M section not the results.
5. Substitute primary or secondary objectives by a description of the objectives as such. For example. Overall safety and safety associated to sensitization profiles
6. Filed stings are a measure of efficacy. I think ordering results in two parts, one related to safety and the other linked to indicating efficacy, might be better to follow the results.
7. It would be interesting to have a table with the characteristics (sensitization profiles) of the 34 patients to understand if the different profiles re covered by AIT in terms of efficacy. A deeper analysis of this part is needed. For example, in the discussion part efficacy data are discussed without a previous description in results (Api m 10 profiles). Discussion is too long compared to M&M and Results.
8. Conclusions. It seems more a marketing message that discussing actual data. Again, the detailed description, composition.,… of the product should be allocated to M&M section.
9. In general the conclusions part is more an opinion that a conclusion of the study presented. For example the treatment duration and design does not allow to conclude on the curative potential, Proteomic characterization of the extract was not described, the study was not designed to prove superiority to other products. Indicating efficacy data can not certify and endorse the efficacy and contradicts the correct statement that more studies are needed. I would simply eliminate all this part or completely rewrite it according to actual presented data. Something as : “Safety and indicating efficacy data, support that xxxx product has a balanced safety and efficacy profile indicated to treat bee venom allergic patients with different sensitization profiles. More studies will be needed to poof the disease modifying potential of the product”
Author Response
Esteemed editor;
First of all thank you for reviewing and finding interesting our article.
General comments:
- "Grammar. The article is difficult to follow. A complete rewriting by an English speaker is strongly recommended. Check homogeneity of verbal tenses. There are multiple grammar errors along the article"
- The manuscript have gone through an extensive grammar review and re-styled for a better understanding.
- Title. "The title should reflect the reality of the study, avoiding “marketing messages”. An alternative title suggestion might be:"
- We really liked this suggestion, thank you in advanced, the manuscript title has been changed for "Real-world safety and efficacy clinical data of an improved Allergen Specific Immunotherapy product for the treatment of bee venom allergy”
- "In general when an original article, sponsored by a Company is submitted, there should be an special care to present scientific evidence without a clear marketing orientation. By doing that, the strength of the publication and lack of bias will be increased and at the end it will be even more positive for the Company purpose. I suggest to revise"
- It has been revised for a more scientific view.
Specific comments:
- Introduction: It has been re-edited.
- Materials and Methods.
- We have added three estructured parragraphs were it is better explained all the material and methods that has been used.
- "A demographic description of the 107 patients included could be incorporated in M&M section describing mean values and SD of the different variables upon inclusion"
- We have included a new table with all the demographic data collected from the clinical histories of the patients.
- "The different protocols used in AIT should be part of the M&M section not the results."
- It has been changed and re-styled
- "Substitute primary or secondary objectives by a description of the objectives as such. For example. Overall safety and safety associated to sensitization profiles"
- We have made the proper changes to clarify the section.
- " Filed stings are a measure of efficacy. I think ordering results in two parts, one related to safety and the other linked to indicating efficacy, might be better to follow the results."
- We included three new sections, overall safety, demographic data related and efficacy.
- " It would be interesting to have a table with the characteristics (sensitization profiles) of the 34 patients to understand if the different profiles re covered by AIT in terms of efficacy. A deeper analysis of this part is needed. For example, in the discussion part efficacy data are discussed without a previous description in results (Api m 10 profiles). Discussion is too long compared to M&M and Results."
- We have added a new table with this information in appendix 2.
- "Conclusions. It seems more a marketing message that discussing actual data. Again, the detailed description, composition.,… of the product should be allocated to M&M section."
- We have changed it as suggestion.
- "n general the conclusions part is more an opinion that a conclusion of the study presented. For example the treatment duration and design does not allow to conclude on the curative potential, Proteomic characterization of the extract was not described, the study was not designed to prove superiority to other products. Indicating efficacy data can not certify and endorse the efficacy and contradicts the correct statement that more studies are needed. I would simply eliminate all this part or completely rewrite it according to actual presented data. Something as : “Safety and indicating efficacy data, support that xxxx product has a balanced safety and efficacy profile indicated to treat bee venom allergic patients with different sensitization profiles. More studies will be needed to poof the disease modifying potential of the product”"
- Real good suggestion, we have applied it.
Round 2
Reviewer 1 Report
Dear Authors,
In my first review I made a general comment on the form of the manuscript: “I recommend that the team of authors take the necessary time in its preparation. It must be adapted to the specific style of Vaccines, for example regarding the list of references to how they are cited in the body of the manuscript. It must be corrected. In general, the text has a lot of room for improvement. It should be structured in paragraphs (not in sentences). The linguistic style used is very difficult to understand.”
Looking to the revised MS, the authors have ignored this suggestion and they have not consulted the style of Vaccines, especially as references are cited, in the main text, and in the final list. Vaccines | Instructions for Authors (mdpi.com) https://www.mdpi.com/journal/vaccines/instructions
L37-39 “Allergy to honeybee venom (HBV) among beekeeper is around 36%, while in the general population the prevalence of HBV allergy is around 15-25%, with a mortality rate about 0.08 per million inhabitants in Spain. (4) (5) (8) (6) (7)”.
In my first review I asked about the reference that establishes 0.08 deaths (per million inhabitants) from bee stings per year in Spain. With the CIE classification, until last year it was not possible to separate between the different hymenopteran insects that caused death (bees, wasps, bumblebees). So, there is no evidence to support that in Spain due to honeybee stings the mortality rate is 0.08. Some are due to bees, others to wasps, bumble bees, hornets. I have reviewed the recent literature on the top: epidemiology of fatalities in Spain due to hornet, wasp, and bee stings over a 20-year period (1999–2018) expressed in terms of annual rates and per million inhabitants, varied from 0.02 to 0.19 [https://www.mdpi.com/968110].
L52-56 “Human Seric Albumin (HSA) is incorporated in vaccines to stabilize the extract and to prevent immunogens from adhering to the walls of the injection vial. However, because HSA is a protein derived from human blood, there is a theoretical risk that it might contain infectious agents which could make it unsafe for humans”. (18) HSA is widely use in venom extracts although it covers some of the allergens, making it difficult to distinguish all of them, but stabilizes the extract making it last longer.”
Firs of all, reference 17 is missing, and the reference 16 in the original submitted MS, is in the references list, but not in the main of the body text of the introduction.
Reference 16 was changued by 18. In the original submitted MS reference 16 was used to tupport the statement. However, we can read as the conclusion of the paper (ref. 16) that: human serum albumin … in vaccines has not been found to be harmful in humans or experimental animals.
Now a new reference is added: Varicella vaccine without human serum albumin versus licensed varicella vaccine in children during the second year of life: a randomized, double-blind, non-inferiority trial. https://doi.o. Prymula, R., Simko, R., Povey, M. et al. 7, 2016, BMC Pediatr, Vol. 16.
In the introduction section of the above paper from Prymula you can read:
“HSA is incorporated in live, attenuated vaccines tostabilize the virus and prevent immunogens from adher-ing to the walls of the injection vial. However, becauseHSA is a protein derived from human blood, there is atheoretical risk that it might contain infectious agents[26] which could make it unsafe for humans.
Reference 26 is again Offit PA, Jew RK. Addressing parents’ concerns: do vaccines contain harmful preservatives, adjuvants, additives, or residuals? Pediatr. 2003;112:1394–7. Again, research carried out by Offit concluded that: human serum albumin … in vaccines has not been found to be harmful in humans or experimental animals.
HSA is commonly used as a stabilizer and excipient in vaccine production. HSA is a safe and well-established component in vaccine formulations and has been used for many decades without any significant adverse effects. HSA is highly purified and undergoes rigorous testing to ensure its safety and efficacy in vaccines. The World Health Organization (WHO) and regulatory agencies around the world have established strict guidelines for the use of HSA in vaccine production. These guidelines ensure that the HSA used in vaccines is free from contaminants and meets stringent quality standards. In summary, the use of HSA in vaccine production is not a significant risk for patients. HSA has a long history of safe use in vaccines, and regulatory agencies closely monitor its use to ensure its safety and efficacy in vaccine formulations. Moreover, just to remember what happened here in USA with the tick-borne encephalitis (TBE) Vaccine (from Pfizer). HSA removal (in 2000) was associated with a substantial increase in the rate of high fever in infants and young children. In 2001 HSA was added again to the TBE vaccine and the incidence of adverse reactions decreased to expected levels.
L70-71 “Seven hospitals in Spain, which were in Pontevedra, Vigo, Lugo, Castellón, Cuenca and Barcelona;
Indicate how many patients per hospital. Moreover, indicate the city with two hospitals, since we can read 7 hospital in 6 cities.
L103-104 “The collection method is by electrical stimulation and without affecting honey production or bee aggressiveness”.
“and without affecting honey production or bee aggressiveness”. Remove the above. Electrical stimulation to harvest venom at the beehive, always increase the honey bee aggressiveness. Provide a reference to support the above statement related to “whithout affecting honey production”.
Author Response
Dear Editor,
Thank you for your kind review, here you can find a new copy of the manuscript and the answers to the questions posed.
Sincerely.
Dr lópez.
- "In my first review I made a general comment on the form of the manuscript: “I recommend that the team of authors take the necessary time in its preparation. It must be adapted to the specific style of Vaccines, for example regarding the list of references to how they are cited in the body of the manuscript. It must be corrected. In general, the text has a lot of room for improvement. It should be structured in paragraphs (not in sentences). The linguistic style used is very difficult to understand.”
Looking to the revised MS, the authors have ignored this suggestion and they have not consulted the style of Vaccines, especially as references are cited, in the main text, and in the final list. Vaccines | Instructions for Authors (mdpi.com) https://www.mdpi.com/journal/vaccines/instructions "
It has been corrected.
2. "L37-39 “Allergy to honeybee venom (HBV) among beekeeper is around 36%, while in the general population the prevalence of HBV allergy is around 15-25%, with a mortality rate about 0.08 per million inhabitants in Spain. (4) (5) (8) (6) (7)”".
It has been corrected and modified
3. " “Human Seric Albumin (HSA) is incorporated in vaccines to stabilize the extract and to prevent immunogens from adhering to the walls of the injection vial. However, because HSA is a protein derived from human blood, there is a theoretical risk that it might contain infectious agents which could make it unsafe for humans”. (18) HSA is widely use in venom extracts although it covers some of the allergens, making it difficult to distinguish all of them, but stabilizes the extract making it last longer.” "
The references have been changed and removed the sentence.
4. "L70-71 “Seven hospitals in Spain, which were in Pontevedra, Vigo, Lugo, Castellón, Cuenca and Barcelona;
Indicate how many patients per hospital. Moreover, indicate the city with two hospitals, since we can read 7 hospital in 6 citie"
One hospital in Tenerife was missing, it has been added also added appendix 1 is the list of the hospital, principal investigator and patients recruited.
5. "L103-104 “The collection method is by electrical stimulation and without affecting honey production or bee aggressiveness”.
“and without affecting honey production or bee aggressiveness”. Remove the above. Electrical stimulation to harvest venom at the beehive, always increase the honey bee aggressiveness. Provide a reference to support the above statement related to “whithout affecting honey production”."
This data hasn't been published it is aprt of a master degree investigation, the name of the investigation is "OPTIMIZATION AND CHARACTERIZATION OF APIS MELLIFERA VENOM FOR APPLICATION IN ALLERGY" by Rosa Gallego Alvarez (Universidad Complutense de Madrid) 2017-2018
were we did all the studies about the bees, behaviour, diffferences between hives and site of collection, different times and cycles of activating the electristimulation device, and the whole study of the venom.

Reviewer 2 Report
No further comment
Author Response
Dear editor;
Thank you for your review.
Round 3
Reviewer 1 Report
L38-39 "with a mortality rate due to hymenoptera stings that varied from 0.02 to 0.19 per million inhabitants in Spain. [4] [5] [6] [7] [8]".
The above statement need to include the original/raw reference:
Feás, X. Human Fatalities Caused by Hornet, Wasp and Bee Stings in Spain: Epidemiology at State and Sub-State Level from 1999 to 2018. Biology 2021, 10, 73. https://doi.org/10.3390/biology10020073
To include in M&M the reference to the work OPTIMIZATION AND CHARACTERIZATION OF APIS MELLIFERA VENOM FOR APPLICATION IN ALLERGY
Author Response
Dear Editor;
Thank you for your kindly review.
We have added the original reference and deleted the one from the book.
Also we have mentioned the master degree work.
Hope this finds you well.
Kind regards.